# Review on Optimization Techniques of PV/Inverter Ratio for Grid-Tie PV Systems

Hazim Imad Hazim [1], Kyairul Azmi Baharin [1,*], Chin Kim Gan [1], Ahmad H. Sabry [2] and Amjad J. Humaidi [3,*]

1 Faculty of Electrical Engineering, Universiti Teknikal Malaysia Melaka, Durian Tunggal 76100, Malaysia
2 Department of Computer Engineering, Al-Nahrain University, Baghdad 64074, Iraq
3 Department of Control and Systems Engineering, University of Technology, Baghdad 10066, Iraq
* Correspondence: kyairulazmi@utem.edu.my (K.A.B.); amjad.j.humaidi@uotechnology.edu.iq (A.J.H.)

**Abstract:** In the literature, there are many different photovoltaic (PV) component sizing methodologies, including the PV/inverter power sizing ratio, recommendations, and third-party field tests. This study presents the state-of-the-art for gathering pertinent global data on the size ratio and provides a novel inverter sizing method. The size ratio has been noted in the literature as playing a significant role in both reducing power clipping and achieving system optimization. The majority of researchers observed that due to varying irradiance distributions and operating temperatures at particular sites, the sizing ratios were dependent on geographic latitude. This study will identify the issue that makes it challenging to acquire dependable and optimum performance for the use of grid-connected PV systems by summarizing the power sizing ratio, related derating factor, and sizing formulae approach. The present study recommends a Deep Learning technique that might, due to the dynamic behavior of the PV technologies, provide fully automatic computation for the DC/AC sizing ratio, and effectively lower the whole return on investment (ROI) over a variety of circumstances and climatic changes.

**Keywords:** optimization; inverter sizing; photovoltaic systems; grid-connected; DC/AC ratio; PV system cost





## 1. Introduction

The use of solar photovoltaic (PV) technology to gather energy from the sun for a variety of purposes around the world offers several benefits, including free and rich sources of energy with an ecological appeal. The instability phenomenon, known as potential-induced degradation (PID) and light-induced degradation (LID), in crystalline PV module technology, however, has been constantly discussed as a weakness and challenge in PV system technologies that have been encountered while sizing and designing. This phenomenon can result in continuous technical hazards on reliability and durability for its actual area act [1,2].

Much effort has been spent to optimize the suitability demands of the inverter and PV array using a precise methodology designed to optimize the grid-connected PV systems [3–7]; however, the problem is still present with certain recently installed methods [8–10]. This matter was overlooked for the sizing of inverters in terms of an LID event, which frequently results in power losses in power clipping occurrences and degrades the performance of the system. Therefore, while working on this type of solar panel technology, the inverter's undersizing of the system by numerous designers and researchers will affect the power inverter and the additional design protection components [11–13].

Inverter sizing for PV systems has been the subject of much research in the literature. In these experiments, the size of the PV inverter was established using one of the two approaches described in [14–18]: (1) it matched the PV array's nominal capacity; and (2) as a general rule, it was undersized at 70% of the PV array's capacity. However, both

approaches fail to take into account crucial elements that determine the PV inverter's ideal size. The ideal size of PV inverters has been determined in further new studies using systematic approaches that take into account a variety of variables, including meteorological circumstances, economic factors, and intrinsic inverter properties [14,19–23]. These studies showed how the inverter loading ratio [24], the levelized price of electricity [25], and PV system installation parameters can all have an impact on the size of the PV inverter that is most appropriate. The term "oversizing ratio" typically refers to the ratio of the inverter's rated AC output power to its maximum DC input power in a controlled testing environment. Oversizing is a crucial metric for assessing the inverter's performance and one of the primary factor installers taken into account when building a PV plant. An illustration of the oversizing and how it affects output power is shown in Figure 1.

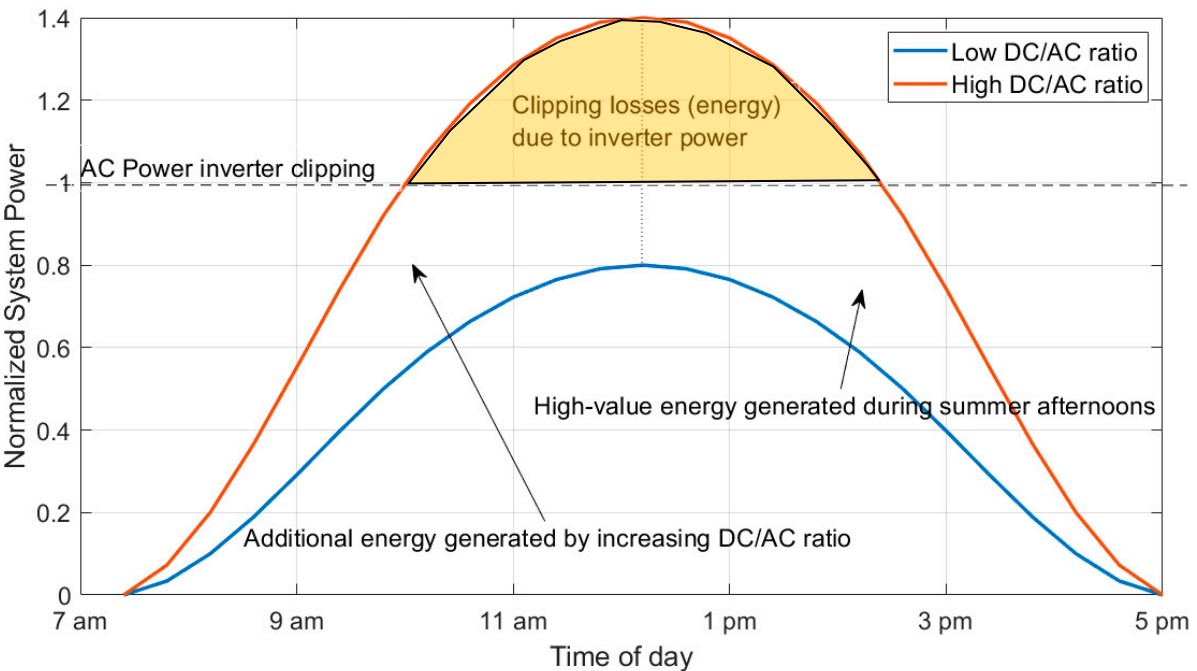

**Figure 1.** Explanation of the oversizing ratio of the DC solar PV-to-inverter AC power output over a whole day.

When there is enough sunlight, the PV array's power output will exceed the inverter's rated maximum output power. At this point, the inverter will restrict the system's current to its maximum rated value, increasing the DC voltage appropriately. In this case, the system output is constrained to the inverter's rated maximum output, and the oversized portion's potential production capacity will be lost. As depicted in the picture, the system's actual power curve will have a flat, straight line for its peak rather than the original normal distribution curve. Peak clipping, or just "clipping," is the name of this procedure.

Taking into account PV surface orientation, inclination, tracking system, inverter characteristics, and insolation, Ref. [26] established the ideal array/inverter sizing ratio for a PV system. The most relative references that mainly discussed the optimization of DC/AC ratio, cost, and tilt angle to maximize annual energy yield for grid-connected PV systems are [18,27–30]. These studies were either based on iterative algorithms or trial-based methods that require a very long time to approach the optimization value of the DC/AC ratio and/or cost. In order to close this gap, this paper empirically analyzes and summarizes the literature on inverter sizing ratios based on the various types of solar PV panel technologies in use worldwide. Moreover, this study focuses on the issues of different PV component sizing methodologies, including the PV/inverter power sizing ratio, and recommendations for PV-inverter systems by summarizing the power sizing ratio, related derating factor, and sizing formulae approaches. In addition, the presented study recommends a Deep

Learning-based technique that might provide fully automatic computation for the DC/AC sizing ratio and effectively lower the whole return on investment (ROI) over a variety of circumstances and climatic changes.

This study also introduces a novel inverter sizing strategy using the Deep Learning network technique that can provide the best value for the sizing ratio. In order to formulate the existing problem, including the sizing ratio frameworks, based on numerous studies, the articles were analytically reviewed by compiling and dividing them from outside sources into several of the most important and diverse topics. This also included analyzing the sizing ratio, which was compiled and divided into several main climatic types according to the Köppen–Geiger climate classification [31], which is freely accessed. A comparison with similar review work is listed in Table 1.

**Table 1.** Comparison with related review studies on DC/AC sizing for PV-inverter systems.

| Ref. | Range of Discussion on DC/AC Sizing and Cost | Literature Focus | Related Analysis and Results | Inverter Undersizing | Proposing System | Date Publish |
|------|------|------|------|------|------|------|
| [32] | Extensive | sizing optimization issues, hybrid PV/wind/diesel generator systems, hybrid PV/wind systems, hybrid PV/diesel generator, and standalone PV systems | No | Limited | No | 2013 |
| [33] | Limited | Power and Energy Losses in PV Plants in Future Ancillary Services Markets | Limited | No | No | 2020 |
| [34] | Limited | Optimization goals, utilized optimization methods, grid type as well as the investigated technology | Yes, statistical results | No | No | 2018 |
| [35] | Limited to PV system installed | Environmental, PV system, installation, cost factors as well as other miscellaneous factors | Limited | No | No | 2017 |
| This work | Extensive | DC/AC ratio optimization techniques | Yes, main results | yes | Yes | 2023 |

## 2. Literature Review

In a grid-tied solar PV system, optimization of DC/AC ratio, cost, and tilt angle to maximize annual energy yield has been discussed and continues as a challenging task for investing in PV systems. A short context of a number of situations over outdoor measurements connected to the impact of the DC/AC ratio towards maximizing the annual energy yield of grid-tied PV systems was addressed in this section.

The study [36] analyzed the optimal use of PV array to inverter sizing for grid-tied systems. In order to determine the ideal grid-connected PV system size, factors such as carbon dioxide ($CO_2$) emissions percentage, net present cost (NPC), the percentage of renewable electricity, excess electricity, and unmet load weretaken into account. It was reported that the DC/AC inverter ratio with a unity value and minimized $CO_2$ emissions produced the best results for providing energy (to Mecca, Saudi Arabia), with excess electricity of 0% and an unmet load. However, it was found that it is possible to downsize

the inverter size to 68% with respect to the nominal PV power to decrease the total NPC of the system, as well as reduce inverter cost.

## 2.1. Derating Factor of PV Technology

The derating factor in PV technology is not difficult to understand from the standpoint of system design concerns. Numerous researchers have remarked that they are more concerned with the number of electrical characteristics produced by PV modules, such as voltage, current, and power outputs. Three researchers stated that the LID phenomenon has a detrimental effect on the usage of PV in terms of sizing consideration. Their study [37] only conducted one investigation, which accounted for a sizing ratio value of approximately 0.98. When designing and sizing, the recommended value should be adjusted between 0.90 and 0.99. However, as DC/AC increases, the inverter is more likely to derate.

The preliminary power stability of PV technologies was confirmed below 1%, while only a few cases showed more than 4%, according to other authors [38]. Over several days of exposure, stability was observed for all of them. One recent study, summarized in [39], revealed that with c-Si technology, the preliminary LID often takes place over the variety of 2% to 4%, with LID predominating in terms of power decrease. According to a study by [40], the potential of the LID phenomena may typically be created by 30% to 10% in the initial few months of outside coverage.

## 2.2. PV Array to Inverter Sizing Strategies

InMalaysia, the typical derating factors for the PV to inverter power size ratios utilized are 1.00 to 1.30 Thin-Film and 0.75 to 0.80 for the c-Si PV type [41]. These calculations takeinto account a variety of variables, including the environment, the mounting structure installation, system applications, PV module technology, type of inverter, and others. The PV-to-inverter ratio ($PR_k$) for derating factor might be determined for the system configuration's dimensioning using Equation (1);

$$PR_k = \frac{P_{ac\_inv}}{P_{PV@STC}} \tag{1}$$

According to [42], the PV-to-inverter power ratios can be expressed by Equation (2) for thin-film PV modules and (3) for crystalline PV modules by:

$$P_{ac\_inv} = 0.889 \times P_{PV@STC} \tag{2}$$

$$P_{ac\_inv} = 0.76 \times P_{PV@STC} \tag{3}$$

Researchers and members of the solar industry created a number of sizing techniques, which were gathered and separated into two approaches. The following is a summary of all the data that was used to determine the optimal plan according to inverter techniques associated with the PV-to-inverter ratio sizing:

- Manufacturers' recommendations based on PV guidelines.
- DC/AC sizing ratio according to third-party publications.

In order to provide an overview of the effect of the ratio of the DC power of PV to inverter AC output power on the cost, the revenue factor was introduced [43]. The normalized revenue ($R_{norm}$) relation can be derived by:

$$R_{norm} = \frac{1 + A\delta}{\left( \frac{y(\delta = \delta_{new})}{y(\delta = 1.0)} \right)} \tag{4}$$

where, according to a solar farm's DC/AC ratio, "A" represents the portion of the total capital invested in the solar farm that has been spent. The $\delta$ is the DC/AC ratio (>1.0) and y denotes the amount of annual yield. Normalized revenue vs. DC/AC ratio at 35° Tilt, 0°

Az, North Victoria/South NSW (35°) with fixed tilt angle is shown in Figure 2 left, while tracking tilt angle in Figure 2 right.

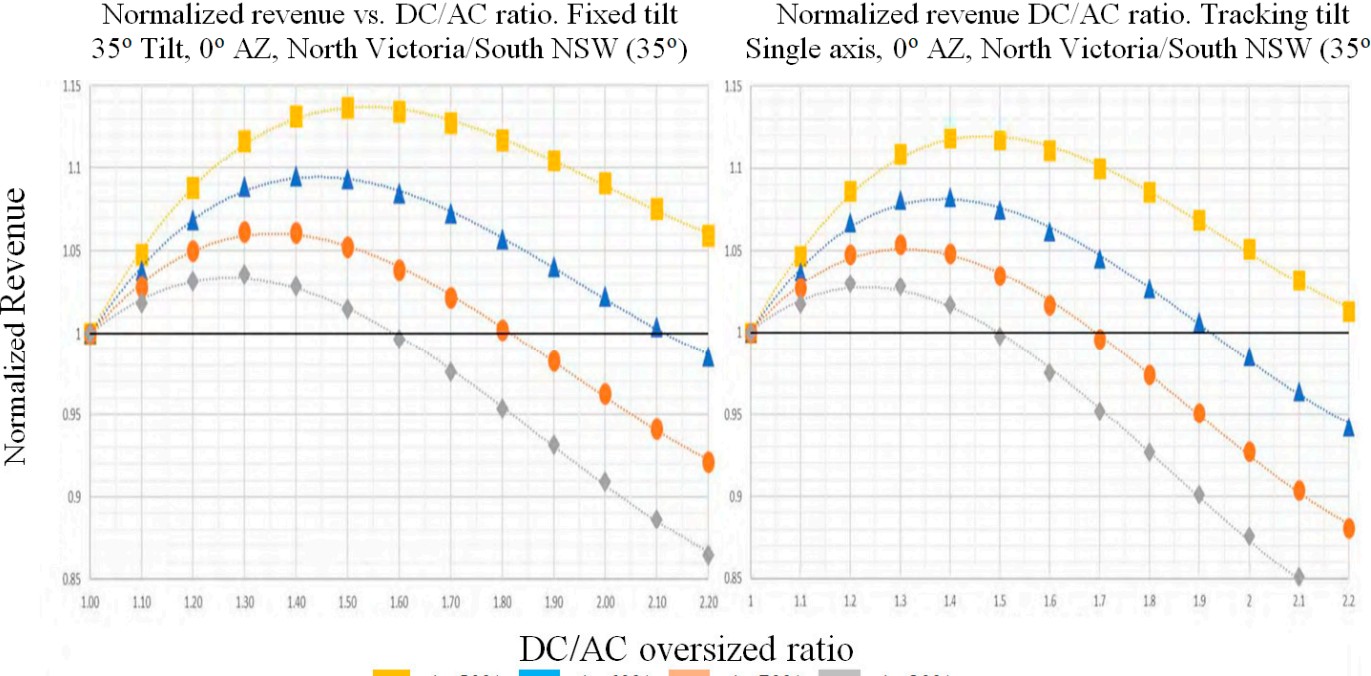

**Figure 2.** Normalized revenue vs. DC/AC ratio at 35° Tilt, 0° Az, North Victoria/South NSW (35°) with fixed tilt angle on the left, while tracking tilt angle to the right.

### 2.2.1. Manufacturers' Recommendations Based on PV Guidelines

The content of this section can be divided into three parts: the first part discusses the guidelines or inverter manufacturers' recommendations based on the PV sizing ratio; the second part, the table, briefly summarizes recommendations of some PV manufacturers and academics as concrete examples in commercial markets; and finally, in the third part, a graphical representation is presented for the chronological summary of the main PV-inverter ratio sizing studies.

As per the DTI Sustainable Energy programs guideline [44,45], the initial few weeks of operation can see increased electrical values. Nominal output power, Isc, and Voc will typically be larger during the initial phase of this operation than every rate determined using a conventional duplication feature. PV arrays can remain unplugged for the initial term to prevent the use of excessively large inverters.

Based on the guidelines of the Clean Energy Council [46], the highest preliminary production of a-Si PV array is typically 25% greater than the power rating, hence the circuit breakers, switches, and inverters all need to be sized to prevent this from happening. This can often take up to six months after installation. Furthermore, the inverter size can be calculated by multiplying the values of the derating factor by the PV array peak power capacity at STC rating, which is reported to be 0.889 for thin film and 0.76 for a-Si-PV technologies, in Australia's guidelines [47]. Three different forms of derating variables, such as temperature, dirt, and manufacturing tolerance, were used to calculate such values for the two different PV panel technologies. If the inverter is designed efficiently, its nominal AC power output might be no less than 75% of the rated PV array power.

The California Energy Commission (CEC) [48] stated that the field-based PTC rating of the input to the inverter output power (PV modules) is recommended as the best practice, and it must include a correction for the initial light-induced degradation. The PTC rating may imply greater performance under field operating conditions, as mandated by the CEC.

Sharp Electronics manufacturer [49] reported that for initial and stable condition values, the system design should be taken into account. Actual initial numbers for power will be greater by approximately15%; the power output is the Staebler–Wronski post-initial decay.

Du Pont Apollo manufacturer [50] recommended that the power surpass the STC value as a state of the industrial qualifications during the stabilization phase, which takes place within the first month of service.

Baoding Tian Wei manufacturer [51] reported that when determining component Isc, Voc, regulator sizes related to the PV power output, fuse sizes, conductor ampacities, and voltage ratings of the datasheet specification should be multiplied by a factor of 1.25. If referring to stable output, a design consideration must account for this variance as an option for the system.

BP Solarex manufacturer [52] recommended that the output of the Millennium product will decline during the first several months of exposure. The rated electrical characteristics of Millennia take the attenuation into account, up to 6% more current, 12% more voltage, and 18% more initial power than specified.

According to Sputnik Engineering inverter manufacturers [41], the distribution of the quantity of irradiation each year ina certain area affects the choice of the dimensioning factor. In order to help their customers design the ideal system, the manufacturer alsocreated a simulation program named Max-Design that is compatible with Solarmax's production. For a broad range of inverter sizing values from 0.80 to 1.10, the adjustment dimensioning factor (DF) may be used according to the specific location in their simulation [53]. However, as larger inverters cost more per watt, the optimal ratio must not be larger than 20% of the power rating of the PV array.

The highest factor "over-dimensioning" of a Solar-Max inverter might be up to 15%, which could lead the PV-rated power to design with 15% more than the chosen AC power capacity of the inverter, according to two university–industry collaboration studies conducted by Danfoss PV Inverters A/S with ISE Germany, Fraunhofer, and Sputnik Engineering. However, the authors recommended that the inverter capacity and PV array power must be rated at 1.0:1.0 ratio as an ideal case. In the second study, B. Burger tested the two types of PV panel technologies to match the inverter Danfoss products with the PV array-rated power in sites around central Europe. The suggested ratio ranged from 1.06 to 1.11 for the Thin-Film PV plant [54].

According to ABB Solar [55], the inverter might be sized between the PV array power and active power of the inverter ratings (0.80 to 0.90). The recommended size ratio considered all power losses that would affect the inverter's power generation and conversion efficiency when it was in use.

A summary of the PV-to-inverter ratio considered in previous studies according to software packages, books' syllabi, and guidelines is listed in Table 2, while a chronological summary of the main related PV-to-inverter sizing ratio approaches is shown in Figure 3.

### 2.2.2. DC/AC Sizing Ratio According to Third-Party Publications

Currently, there is disagreement among PV specialists over whether it is economically feasible to create an optimization system. The "rule-of-thumb" approach has been cited by several academics as an effective way to determine the ideal inverter power ratings and PV array arrangement. Achieving the ideal size in their systems was explained in detail by other researchers as well. System design issues are becoming more crucial as optimization approaches ensure that the operating system functions optimally, reliably and with excellent conditions. As a result, system integrators must give careful consideration to balance-of-system (BOS) component selection inthe original design phase.

**Table 2.** Summary of PV-to-inverter sizing ratio based on the country and the recommendation.

| Ref. | PV/Inverter Ratio | Company/Country | Recommendations |
|---|---|---|---|
| [56] | 0.88–1.1 | KACO New Energy | Power Ratio = $PV_{GEN}/P_{AC,INV}$ |
| [57] | 0.7–1.0 | Power-One Inc. | PV Power @ STC/AC Power Nom. Max. of Inverter |
| [58] | 1.0 | Leonic Co., Ltd. | N/A |
| [54] | 0.8–1.2 | Danfoss Solar Inverters | Si PV = 0.94; Thin-Film = 0.94–0.90 and Thin–Film = 1.0 if Free-standing |
| [59] | 0.75–0.85 | AE PV-powered Inc | N/A |
| [60] | 0.8–1.2 | SMA Solar AG | PV/inverter power ratio (Vp) = input power inverter/peak power PV (0.9–1.0); Accepted Vp = 0.8–1.2 = (under extreme climate) |
| [61] | 0.8–1.1 | Energy, Staffelstein & Engineering | DF (Dimensioning factor) = $P_{solar}/P_{WR,ACmax}$ < 0.8: for DF = 0.8–1.15 = inverter too high; recommended for 35° inclination and south orientation; DF = (1.2–1.3): recommended facades (90° inclination), west or facing east; DF over 1.3: inverter too small; DF = (1.15 to 1.2): recommended to orient well to a very flat module under 15° inclinations or/and south (SW, SE). |
| [62] | 1.3–0.8 | Solar Photovoltaic Power: Designing Grid-Connected Systems, Malaysia | $PR_k = \frac{P_{ac\_inv}}{P_{PV@STC}}$ For Si PV = 0.80–0.75; for Thin–Film = 1.30–1.00 |
| [63] | 0.7–1.5 | UD, Delaware, US, Syllabus Book | Cost-effective and limited choice of inverter sizes to choose SF, even if overloaded occasionally. |
| [64] | 0.7–1.0 | Europe | Southern Europe (35–45° N) = 1.0–0.85; Central Europe (45–55° N) = 0.9–0.75; Northern Europe (55–70° N) = 0.8–0.7; |
| [65] | 0.8–1.2 | India | N/A |
| [66] | 0.7–0.65 | United States | N/A |
| [45] | 1–0.8 | United Kingdom | PV array-to-inverter ratio must be sized between 1:0.8 to 1:1 |
| [67] | 0.75 | Guideline/Standard Australia | The nominal AC output power of the inverter cannot be under 75% of the peak power of the PV array. |

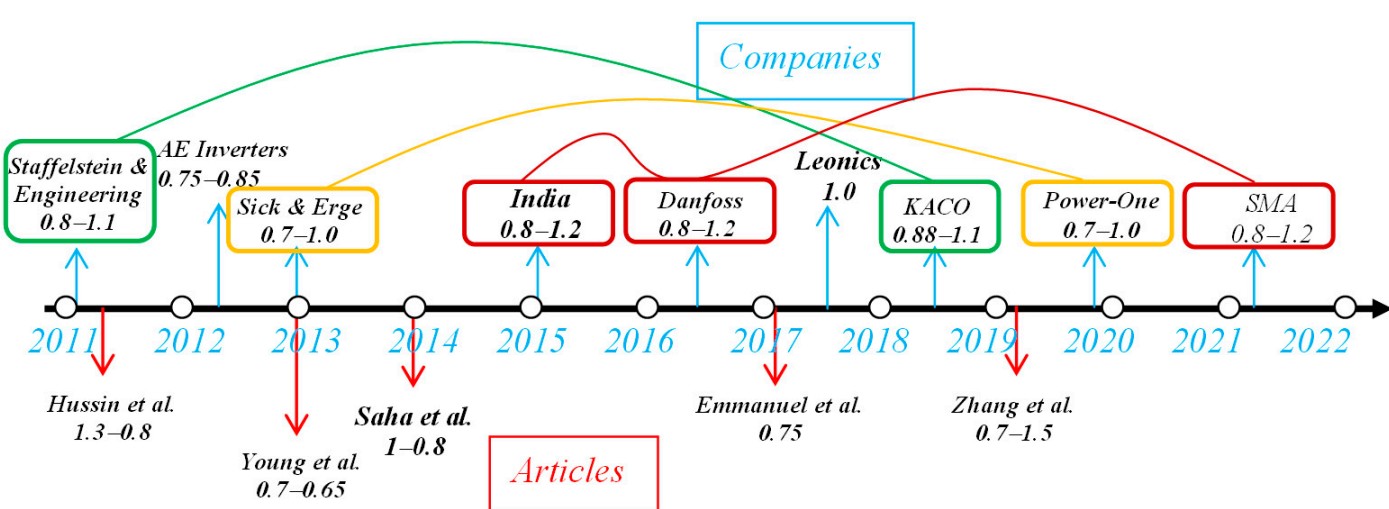

**Figure 3.** Chronological summary of the main PV/inverter ratio sizing related approaches.

In order to maximize the amount of energy injected into the grid, it is vital to combine inverter and PV array components for a grid-tied PV system in order to obtain the ideal size ratio. The optimal sizing ratio, according to Burger et al. [15], relies on the geographic location characteristics, the PV inverter, and the module material composition. To reduce the influence of the chosen inverter size on calculations and to prevent errors in power distribution, the study recommended that the precision of the measured time interval must not be five minutes or less. In contrast, the recommended size ratio took into consideration all power losses that would affect the inverter's power generation and conversion efficiency when it was in use.

There are benefits and drawbacks to both undersized and large inverters according to size in relation to the rated power of the PV array. Consequently, throughout the maximum irradiance, where the inverter that clips power could allow the inverter components to overheat, an undersized inverter would directly affect energy generation. Alternatively, higher power inverters will operate at reduced efficiencies, particularly in medium and low irradiance steps, which results in increasing inverter costs and limits the system's potential to make money. The grid-connected system performance is significantly impacted by the choice of the inverter, which may be either oversized or underpowered in relation to the STC power capacity of the PV array, as stated in [68].

According to [37], the ideal PV/inverter size ratio is significantly influenced by factors such as inverter efficiency, orientation and inclination, local climate, and location. In addition, the authors recommended that the capital expenses of the PV-to-inverter cost ratio (T) be taken into account when forming the top sizing ratio value. For the highest inverter efficiency, the ratio of inverter sizing (Rs) must be scaled within 1.3–1.4 (low irradiation) and 1.1–1.2 (high irradiation) in specific European locations, such as Nancy, Stuttgart, London, Almeria, and Madrid. The Rs value must be estimated in the range of 1.4–1.5 on low irradiance and 1.2–1.3 on high irradiance when dealing with low inverter efficiency [69]. The Rs is determined in terms of the inverter sizing ratio ($P_{inv,\ rated}$) (dimensionless) and the array capacity at STC rating ($P_{PV,\ rated}$) by Equation (4) [70];

$$Rs = \frac{P_{PV,rated}}{P_{inv,rated}} \tag{5}$$

In order to determine the ideal inverter downsizing coefficient, the study [14] suggested a sizing method in the range of $P_{inv,dc,nom,}$ and $P_{PV,nom}$. The authors point out that a number of variables, including weather, price, and inverter features, may impact an inverter scaling strategy. The threshold (occurrence percentage) of irradiance ($G_{TH}$) at a specific site should not be exceeded by the distribution of irradiance, as this may result in excessive power that exceeds the inverter capacity. This could result in some energy losses under greater irradiance and a reduced coefficient of power with temperature. The relationship between $P_{inv,dc,nom,}$ and $P_{PV,nom}$ is called the inverter downsize coefficient (R) and is derived by [71]:

$$R = \frac{P_{inv,\ dc,nom}}{P_{PV,\ nom}} = \frac{G_{TH}}{G_{STC}}; \begin{cases} \text{Undersized inverter for } 0 < R < 1 \\ \qquad\quad \text{oversized for } R > 1 \end{cases} \tag{6}$$

where $G_{STC} = 1000\ W/m^2$ is the STC irradiance, $G_{TH}$ is the irradiance threshold at a chosen site ($W/m^2$), $P_{PV,nom}$ is the rated PV installed power (kWp), and $P_{inv,dc,nom}$ is the DC input rated power of inverter (kW).

Stetz et al. [72] stated that the present performance of the PV-to-inverter sizing ratio, which is given by (P_PV@STC/PWR_AC@NOM) varies between 0.95 to 0.85 in most PV designs operating within a PV plant site in Germany, Bavaria Southeast.

### 2.2.3. A Climate Classification

This work analyzes the sizing ratio and is divided and compiled into several main climatic types based on the Köppen–Geiger climate classification to formulate the current problems based on a number of investigations, as shown in Table 3.

**Table 3.** Climate classifications and their weather descriptions.

| Climate Classification | Country/Territory with the Weather |
|---|---|
| Dfb | Humid continental climate, warm summer; at least four months averaging above 10 °C, all months with average temperatures below 22 °C, and coldest month averaging below 0 °C (or −3 °C). |
| Csb | Mediterranean climate, warm summer; the driest month of summer receives less than 40 mm, at least three times as much precipitation in the wettest month of winter as in the driest month of summer, all months with average temperatures below 22 °C, at least four months averaging above 10 °C, and coldest month averaging above 0 °C (or −3 °C). |
| Csa | Mediterranean climate, hot summer; the driest month of summer receives less than 40 mm, at least three times as much precipitation in the wettest month of winter as in the driest month of summer, at least four months averaging above 10 °C, at least one month's average temperature above 22 °C, and coldest month averaging above 0 °C (or −3 °C). |
| Cfb | Subtropical highland climate or temperate oceanic climate; at least four months averaging above 10 °C, all months with average temperatures below 22 °C, and coldest month averaging above 0 °C (or −3 °C). |
| Cfa | No dry months in the summer. No significant precipitation difference between seasons. Humid subtropical climate; at least four months averaging above 10 °C (50 °F), at least one month's average temperature above 22 °C (71.6 °F),and coldest month averaging above 0 °C (32 °F) (or −3 °C (27 °F)). |
| BSk | Cold semi-arid climate |
| BWh | The hot desert climate, and no month with an average temperature greater than 10 °C. |
| Cwa | Monsoon-influenced humid subtropical climate; at least ten times as much rain in the wettest month of summer as in the driest month of winter, at least four months averaging above 10 °C, at least one month's average temperature above 22 °C, and coldest month averaging above 0 °C (or −3 °C). |
| Af | The average precipitation of at least 60 mm every month (tropical rainforest climate) |
| Aw | The driest month has a precipitation of less than 60 mm (tropical savanna or dry and wet climate). |

The stipulation of reactive and active powers of the presented economic cost was also considered in relation to the systematic technique in calculating the best sizing of the inverter. Additionally, as shown in Table 4, other field studies were conducted in relation to the size ratio theory and formulated values considered in publications by third parties worldwide divided into climatic regions.

**Table 4.** Studies in relation to the sizing ratio theory and values considered in publications by third parties worldwide divided into climatic regions.

| Optimal Power Ratio | Method/Relation | Recommendation | Climate Classification | Country/Group | Ref. |
|---|---|---|---|---|---|
| 1.50–1.00 | $\frac{P_{pv}}{P_{inv}}$ | SI; r = 1.5 medium efficiency inverter, r = 1.2 high-efficiency inverter. HSI; r = 1.10 medium and low-efficiency inverter, r = 1.00 high and medium efficiency inverter. | Dfb | Finland | [73,74] |
| 0.71 | $\frac{P_{inv,dc,nom}}{P_{PV,nom}} = \frac{G_{TH}}{G_{STC}}$ | 0.71 | Csb | Eugene, OR, USA | [14] |

**Table 4.** *Cont.*

| Optimal Power Ratio | Method/Relation | Recommendation | Climate Classification | Country/Group | Ref. |
|---|---|---|---|---|---|
| 0.71 | $\dfrac{P_{inv,dc,nom}}{P_{PV,nom}} = \dfrac{G_{TH}}{G_{STC}}$ | 0.71 | Csa | Sacramento, CA, USA | [14] |
| 1.291–1.204 | $\dfrac{P_{pv,rated}}{P_{inv,rated}}$ | β = 60° (1.204), β = 45° (1.291) | Csa | Batna, Algeria | [75,76] |
| 1.220–1.153 | $\dfrac{P_{pv,rated}}{P_{inv,rated}}$ | β = 60° (1.153), β = 45° (1.220) | Csa | Algiers, Algeria | [77] |
| 0.67 | NA | 0.67 | Csa | Portugal | [78,79] |
| 1.00–0.80 | $\dfrac{P_{max,\,inverter}}{P_{nom,generator}}$ | 0.85 | Cfb | Bogota, Colombia | [80] |
| 0.65 | NA | 0.65 | Cfb | The Netherlands | [81] |
| 1.20–0.75 | $\dfrac{P_{pv-inv,\,nom}}{P_{pv\,peak}}$ | v = 0.90 (Germany) | Cfb | Germany | [82] |
| 0.95–0.85 | $\dfrac{P_{pv}}{P_{inv,\,AC\_nom}}$ | NA | Cfb | Freiburg, Germany | [83] |
| 1.30–1.15 | $\dfrac{P_{max,\,inverter}}{P_{peal\_PV\_array}}$ | 1.15 | Cfb | Nottingham, UK | [15] |
| 0.90–0.70 | $\dfrac{P_{pv,rated}}{P_{inv,rated}}$ | TF = 1.3, Overcast sky = 0.9–0.7 | Cfb | Northern Ireland, UK | [84] |
| 1.10–1.50 | $\dfrac{P_{DC\_STC}}{P_{RATED}}$ | Low Eff. Inv; LSI = 1.4–1.5; HIS = 1.2–1.3, High Eff. Inv; LSI = 1.3–1.4; HIS = 1.1–1.2, | Cfb | Loughborough, UK | [7] |
| 1.25 | $\dfrac{P_{inv,\,dc,\,nom}}{P_{PV,\,nom}} = \dfrac{G_{TH}}{G_{STC}}$ | 1.10–1.40 | Cfb | Oak ridge, TN, USA | [85] |
| 1.25 | $\dfrac{P_{pv,rated}}{P_{inv,rated}}$ | 1.10–1.40 | Cfb | Northern Ireland, UK | [7] |
| 1.25 | $\dfrac{P_{DC\_STC}}{P_{RATED}}$ | TF = 1.10–1.15 | Cfb | Loughborough, UK | [86] |
| NA | $\dfrac{P_{inv,\,dc,\,nom}}{P_{PV,\,nom}} = \dfrac{G_{TH}}{G_{STC}}$ | 0.69 | Cfa | Oak ridge, TN, USA | [14] |
| 1.30–1.20 | $\dfrac{P_{pv}}{P_{inv,\,AC\_nom}}$ | Si PV = 1.30–1.20; Thin-Film < 1.00 | Cfa | UFSC, Florianópolis, South Brazil | [15] |
| 0.83–0.78 | $\dfrac{P_{inv,\,AC\_nom}}{P_{array,\,STC}}$ | Thin-Film Fall = 0.82; Thin-Film Summer = 0.83; Thin-Film Spring = 0.82; Thin-Film Winter = 0.78; | BSk | Golden, Colorado | [87] |
| 1.00–0.60 | $\dfrac{P_{inverter,\,max,\,AC\,output}}{P_{DC,\,rating}}$ | 1.22 | BSk | San Diego, California | [18,27] |
| NA | $\dfrac{P_{inv,\,dc,\,nom}}{P_{PV,\,nom}} = \dfrac{G_{TH}}{G_{STC}}$ | 0.74 | BSk | Prewitt, NM, USA | [14] |
| 0.85–0.65 | $\dfrac{P_{inv}}{P_{array(Act)}}$ | Sfmin = 0.65; Sfmax = 0.85 for Gulf Council Countries | BWh | Kuwait | [88] |
| NA | $\dfrac{P_{inv,\,dc,\,nom}}{P_{PV,\,nom}} = \dfrac{G_{TH}}{G_{STC}}$ | 0.67 | BWh | Phoenix, AZ, USA | [14] |
| NA | $\dfrac{P_{inv,\,dc,\,nom}}{P_{PV,\,nom}} = \dfrac{G_{TH}}{G_{STC}}$ | 1.00 | BWh | Las Vegas, NV, USA | [14] |
| 1.02–0.55 | $\dfrac{P_{inv}}{P_{PV}}$ | NA | Cwa | Sao Paulo, Brazil | [89] |
| 1.321–1.210 | $\dfrac{P_{pv,rated}}{P_{inv,rated}}$ | β = 45° (1.321), β = 60° (1.210) | BWh | Adrar, Algeria | [3] |
| 0.85–1.07 | | Valid on all PV technologies | Af | Malaysia | # |
| NA | $\dfrac{P_{inv,\,AC\_nom}}{P_{PV,\,dc,\,STC}}$ | 0.761 (Lanai)/0.741 (Oahu) | Aw | Lanai/Oahu, Hawaii, USA | [14] |
| 1.43–1.21 | $\dfrac{P_{pv,rated}}{P_{inv,rated}}$ | Valid on all PV technologies | Af | Kuala Lumpur, Kuching and Alor Setar, Johor Bharu, Ipoh, Malaysia | [90] |
| 1.03–0.93 | $\dfrac{P_{inv,max}}{P_{PVG,stc}}$ | Integrated (0.93), Flat surface (1.03) | Csa | Cadiz, Spain | [80] |

r—Sizing ratio; DF—Dimensioning factor, TF—Thin-Film; β—Tilt angle; ISF—Inverter sizing factor; IPR—Inverter power ratio, v—Nominal power ratio, HSI—High solar irradiation; LSI—Low solar irradiation, # Experimental result.

Some articles were accounted for in the classifications starting with A, but the majority was concentrated in classifications B and C, as shown in the Table 4 summary. Regarding the theoretical equation considered to calculate the size ratio of the PV array-to-inverter rated power, there are a number of perplexing approaches. Several publications used different brand names for the pairing of the PV array and inverter, including v, which is the nominal ratio power [91], IPR (inverter power ratio) [92], r (PV/inverter sizing ratio) [93], DF (dimensioning factor) [72], and others.

The system size ratio formula in the literature expressed the power ratio as inverter/PV array and PV/inverter. Regarding the STC PV power capacity, the majority of studies assessed the sizing ratio of the inverter rating power to be between 10% and 40% [72,91,94]. The value of the system size ratio is dependent on how much the tilted PV array's orientation fluctuates, and it tends to rise as the installed PV array's tilt angle rises, according to other authors' summaries, including the orientation of the tilted PV array factor described in [3]'s conclusion. Mondol et al. [69] and Peippo et al. [95] explored the effect toward size ratio according to the different inverter transfer efficiencies from the viewpoint of the used inverter. Both authors came to the conclusion that the use of higher inverter efficiencies caused the sizing ratio to be nearer to 1.0, particularly in high irradiance.

### 2.3. Analytical Methods Affect the Inverter in the PV Inverter

The study by [96] discussed the issues affecting the distribution system as a result of PV penetration, such as harmonics, voltage balance, voltage rise, and voltage fluctuation and their consequences on the system. However, this study did not discuss the PV/inverter power sizing ratio.

Although one study [32] reviewed the sizing optimization issues of PV systems and took into account grid connected systems, hybrid PV/wind/diesel generator systems, hybrid PV/wind systems, hybrid PV/diesel generator systems, as well as standalone PV systems, this study did not discuss the sizing optimization problems for inverter oversizing. In addition, the study has become somewhat outdated on the inverter sizing ratio technology, since it reviewed over 100 articles in the period of 1982–2012.

Analytical studies such as [97] calculated the optimum inverter size in grid-tie PV systems, but with limited (four) unidentified parameters, one related to the location, and three related to the inverter. In the same context, the optimal inverter size for PV systems placed on two-axis tracking mechanisms in European locations ware estimated analytically in [98]. The duration curve of the power of the PV array's DC terminals was used as the foundation for the analytical formulation of the ideal inverter size. However, inverter undersizing issues and inverter clipping were been taken into consideration and the calculations were constrained by the inverter's maximum output power.

Overvoltage issues are frequently brought on by the growing use of PV systems in distribution networks. The provision of reactive power (RP) by the PV converters is one approach to solving this problem. As a result, increased power losses on the PV converters could raise operating expenses. This issue was discussed by [99], where the losses were individually computed for the system inverter as the losses affected by the RP. These losses are comprehensively reviewed in Section 2.2 of [33].

## 3. Recommended Deep Learning for Inverter Sizing

### 3.1. System Cost Consideration

This study recommends running the optimization process of PV array-to-inverter ratio with one platform approach using Deep Learning algorithms taking into account the PV system's whole cost, annual energy yield, and hyperparameters to control the learning process. Therefore, it is essential to identify some terms in this concern such as the net present value (NPV) [100–102], which is determinedby:

$$\text{NPV}(i, N) = \sum_{t=0}^{N} \frac{R_t}{(1+i)^t} \tag{7}$$

where R$_t$ represents the net cash flow at time t. The net present value can be defined as the summation of a time series of cash flows brought into the present. The choice of the ideal PV-inverter ratio that maximizes NPV is a moving target, as represented in Figure 4, while the significant variables that affect NPV and how they interact is shown in Figure 5.

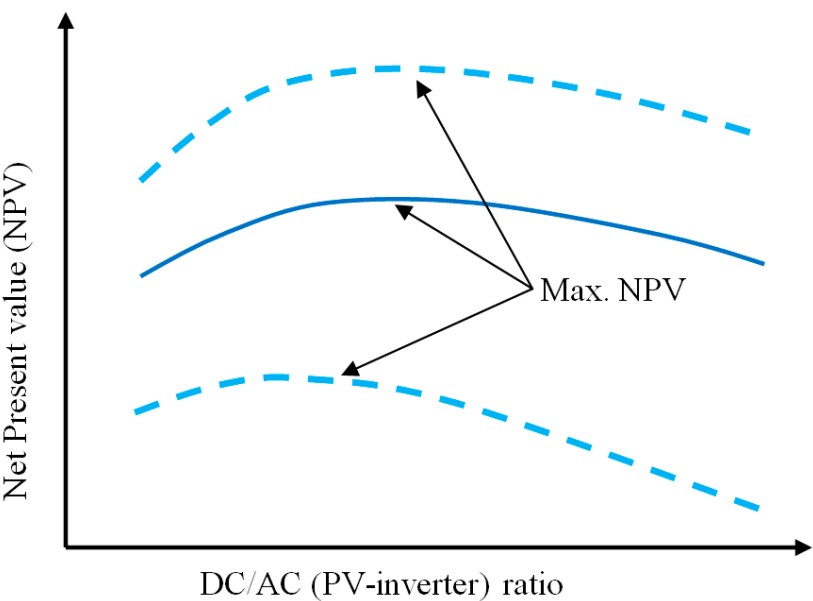

**Figure 4.** Maximizing NPV versus PV-inverter ratio.

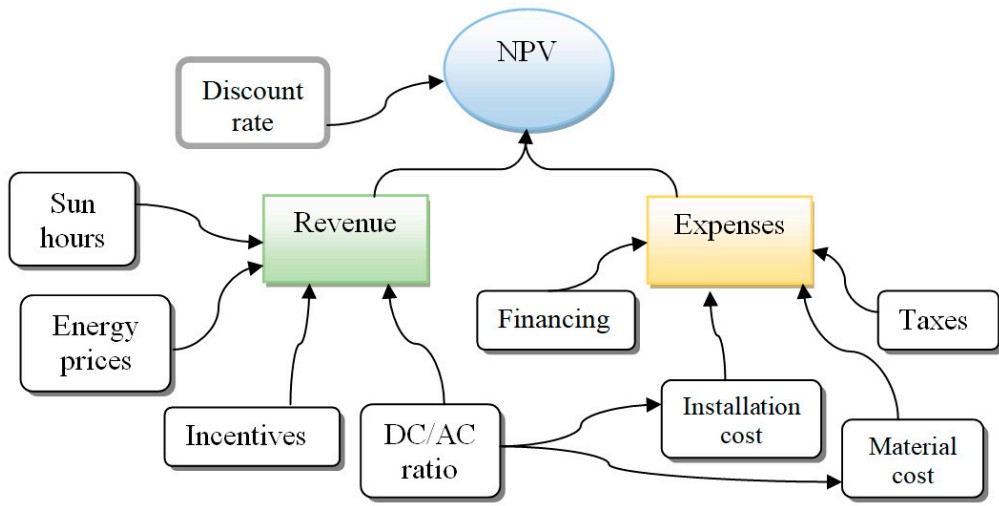

**Figure 5.** The significant variables that affect NPV and how they interact.

### 3.2. Recommended Approach

Numerous studies have been conducted on size, enabling the selection of the best PV Panel (PVP)-battery source in accordance with the loads to be used and the methods for controlling and optimizing the entire system [103–107].Different optimization studies using Deep Learning algorithms for PV component sizing are found in the literature. For example, an energy storage system sizing scheme and PV-dependent navigation routing [108], a sizing algorithm method of a standalone photovoltaic system for powering a mobile network base station [109], and a method for optimal operation and design for PV hybrid plants with battery storage systems [110]. A new approach to annually optimizing energy yield for on-grid photovoltaic systems that use Deep Learning networks is provided in this study, which includes: (1) a new model to compute the yearly PV average energy yield including power conversion stages, system power losses, DC/AC ratio, and overall system costs; (2) a

hysteresis control strategy that guarantees a lower cost with respect to the obtained annually gained PV yields; and (3) design of the solar PV system must consider three key parameters (DC/AC ratio, cost, and PV orientation) that need to be optimized simultaneously. The aim is to maximize the yearly energy yield of the PV system while minimizing its losses and costs under PV site constraints. A diagram of the recommended approach to optimize PV array DC/AC inverter power, while maximizing yearly energy yield for on-grid photovoltaic systems that use Deep Learning networks, is shown in Figure 6.

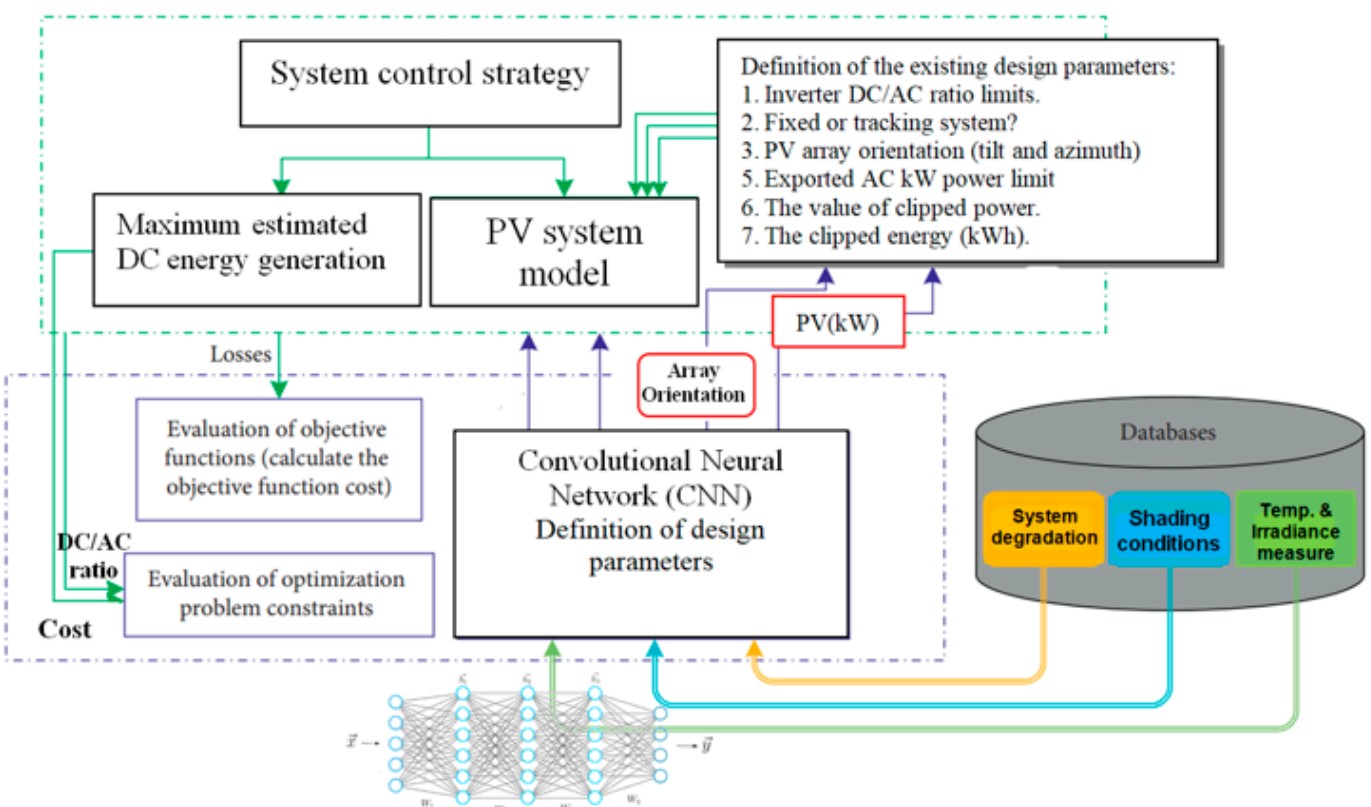

**Figure 6.** Diagram of a recommended approach to optimize PV array DC/AC inverter power, while maximizing yearly energy yield for on-grid photovoltaic systems that use Deep Learning networks.

A historical dataset for the considered PV sites, such as components degradation, shading conditions, and weather measurements, is essential to estimate the yearly energy yield. Convolutional neural network (CNN) algorithms are appropriate in such cases where discrete design variables are used to search for optimal yearly energy yield. It performs a systematic and efficient search among the developed databases for a set of components that define the optimal PV system DC/AC ratio and system costs. The presented approach can help make the task of designing such systems easier, since the yearly yield optimization depends on site conditions and restrictions, component specifications, and the PV array orientation. This will significantly reduce the time to estimate the yearly energy yield.

### 3.3. Results

In this study, a system with a range of 1–5 kWp solar power capacity and an inverter of 2 kWp installed at longitude 44°28′ E and latitude 33°14′ N were considered. The system cost and power records were obtained with the aid of the system advisor model (SAM) [111]. According to [112], the annual weather temperature and irradiance records of Baghdad, Iraq-can be depicted in Figure 7.

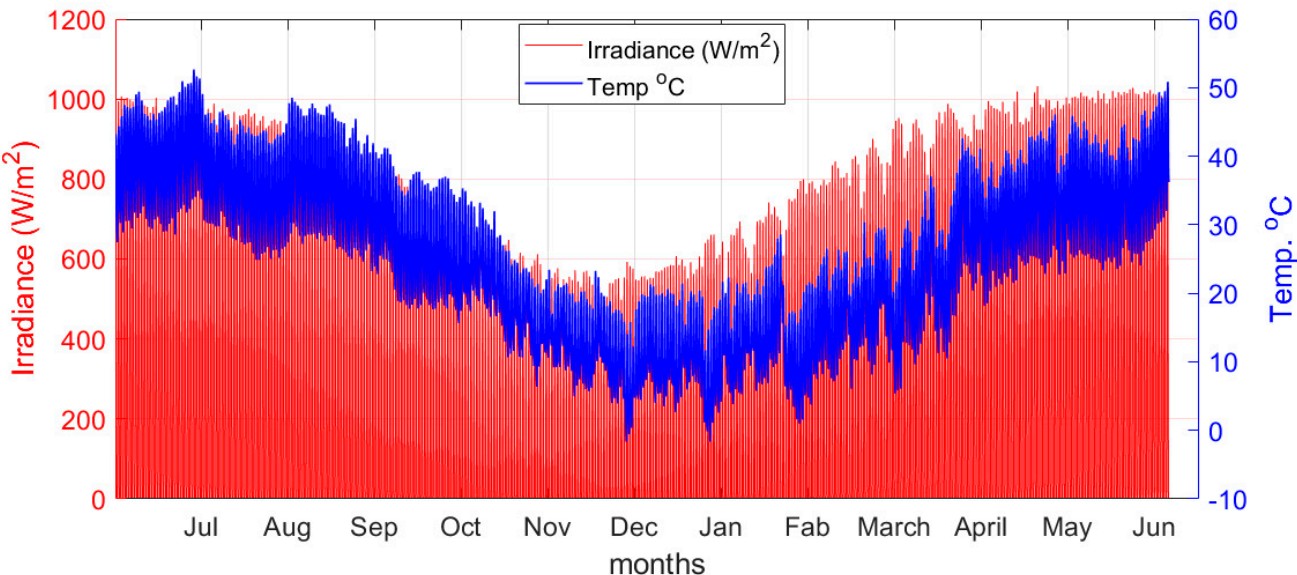

**Figure 7.** The annual weather temperature and irradiance records of Baghdad, Iraq.

The observations show that the irradiance-to-temperature ratio does not remain consistent throughout the year. In addition, there is a difference in the ratio of irradiance to temperature between the year's growth and deterioration halves. To verify the proposed approach, we employed Bayesian optimization CNN with the main specifications listed in Table 5, while the main interesting result showing the relationship between the system cost with respect to PV capacity and the DC/AC ratio for this system is shown in Figure 8.

**Table 5.** List of the main specifications of the proposed Bayesian optimization architecture.

| Description | Dimensions |
| --- | --- |
| Minimum Batch Size | 128 |
| Initial Learning Rate | 0.0003 |
| Maximum Epochs | 15 |
| layers convolution 2d Layer 3 | 3 |
| batch Normalization Layer | 1 |
| relu Layer | 1 |
| Maximum Pooling 2d Layer 3, Stride = 2 | 3, 2 |
| convolution 2d Layer 3, 2 × Number of filters | 3, 2 × 12 |
| batch Normalization Layer | 1 |
| relu Layer | 1 |
| maximum Pooling 2d Layer 3, Stride = 2 | 3, 2 |
| convolution 2d Layer 3, 4 × Number of filters | 3, 4 × 12 |
| batch Normalization Layer | 1 |
| relu Layer | 1 |
| Maximum Pooling 2d Layer 3, Stride = 2 | 3, 2 |
| convolution 2d Layer 3, 4 × Number of filters | 3, 4 × 12 |
| batch Normalization Layer | 1 |
| relu Layer | 1 |
| convolution 2d Layer 3, 4 × Number of filters | 3, 4 × 12 |
| batch Normalization Layer | 1 |
| relu Layer | 1 |
| Maximum Pooling 2d Layer (time Pool Size 1) | 1 |
| dropout Layer | 1 |
| fully Connected Layer (12 = numClasses) | 12 |
| Soft-max Layer | 1 |
| classification Layer | 1 |

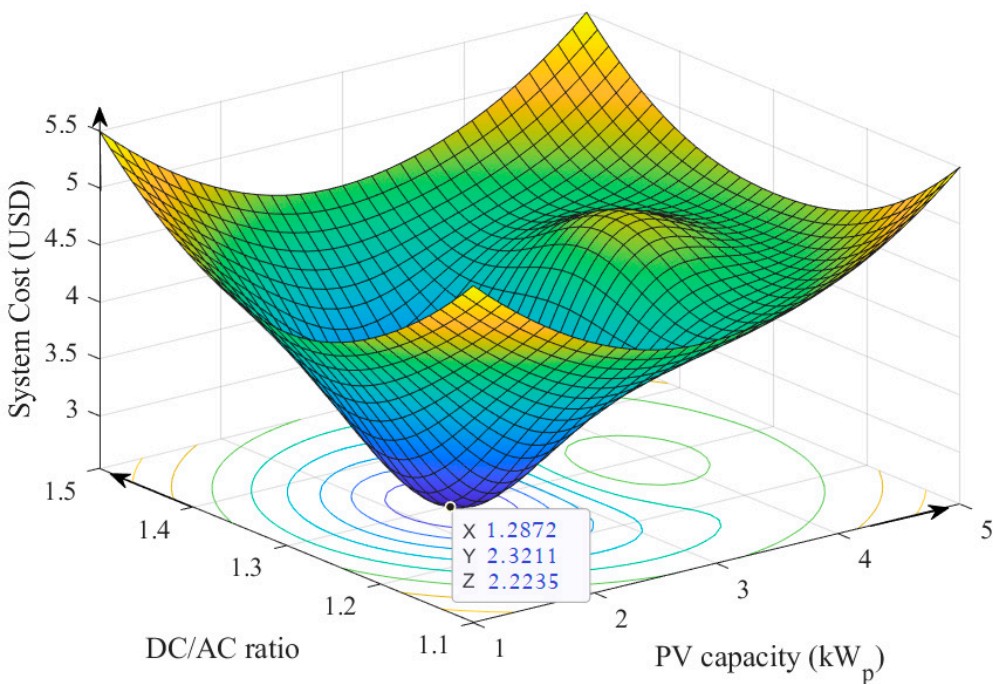

**Figure 8.** The relationship between the system cost with respect to PV capacity and the DC/AC ratio for the studied system.

The obtained results demonstrate that there is a global minimal point for a particular system of specified environmental conditions when minimizing the cost of a system with a 2 kW inverter. Moreover, a minimal cost of 2223 USD was obtained associated with an optimal DC/AC system ratio of 1.287 and a required PV capacity of 2.32 kWp.

## 4. Conclusions

Many studies have discussed the optimization of the PV inverter sizing issue for grid-connected PV systems. The frequently employed inverter-to-PV array formula uses power as a design factor of scaling ratios, and the majority of the studies concentrate on the best uses of c-Si PV module technology. Most studies indicated that the optimal sizing ratio relies on the geographic location characteristics, the PV inverter, and the module material composition, and the recommended size ratio took into consideration all power losses that would affect the inverter's power generation and conversion efficiency when in use.

The most relative references that mainly discussed the optimization of DC/AC ratio, cost, and tilt angle to maximize annual energy yield for grid-connected PV systems are [18,27–30]. These studies were either based on iterative algorithms or trial-based methods that consume a very long time to approach the optimization value of the DC/AC ratio and/or cost. In order to close this gap, this paper empirically analyzed and summarized the literature on inverter sizing ratios according to the various PV module technology types in use worldwide. It also introduced a novel inverter sizing strategy using the Deep Learning network technique that can provide the best value for the sizing ratio. The obtained results demonstrated that under specified climate conditions and component constraints, there is a global minimal point for the cost of a particular PV inverter system. Furthermore, cost minimization was conducted to obtain the corresponding optimal DC/AC ratio and the required PV capacity.

**Author Contributions:** Conceptualization, H.I.H. and A.H.S.; methodology, H.I.H.; software, H.I.H.; validation, K.A.B., A.H.S. and A.J.H.; formal analysis, C.K.G.; investigation, C.K.G.; resources, C.K.G.; data curation, K.A.B.; writing—original draft preparation, A.J.H.; writing—review and editing, A.H.S.; visualization, C.K.G.; supervision, K.A.B.; project administration, H.I.H.; funding acquisition, A.J.H. All authors have read and agreed to the published version of the manuscript.

**Funding:** This research received no external funding.

**Institutional Review Board Statement:** Not applicable.

**Informed Consent Statement:** Not applicable.

**Data Availability Statement:** Not applicable.

**Acknowledgments:** The authors acknowledge the support given by the Faculty of Electrical Engineering, Universiti Teknikal Malaysia Melaka.

**Conflicts of Interest:** The authors declare no conflict of interest.

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
