# Peer review of "Review on Optimization Techniques of PV/Inverter Ratio for Grid-Tie PV Systems"

_applsci, doi:10.3390/app13053155_

Round 1

Reviewer 1 Report

1. The review is not accurate, first of all, in terminology. It is not clear what devide by what, and the fact that DC/AC ratio is not the efficiency of the inverter, it became clear far from immediately. Only page 9 lists several names without definitions, what they mean and how they are related to each other. It seems to me that the article should begin with this.
2. Equations (2) and (3) conflict to the reference to them.
3. In section 2.2.1, half of the description is in text, half is in a table. The principle of separation is not clear.
4. Where did the optimal values come from in Table 2 and why they differ from the recommended ones is not clear.
5. Section 3 is a kind of "protocol of intent" and contains neither the results nor the analysis and characteristics of the initial data with the procedures for their selection, which is a significant part of machine learning methods.
6. The conclusions are abstract, they do not contain useful information.

Author Response

  1. The review is not accurate, first of all, in terminology. It is not clear what devide by what, and the fact that DC/AC ratio is not the efficiency of the inverter, it became clear far from immediately. Only page 9 lists several names without definitions, what they mean and how they are related to each other. It seems to me that the article should begin with this.

 The authors completely agreed with the reviewer’s comment about the clarification of the DC/AC terminology. The reason for this confusion is that various expressions for the relation between the input PV power and the inverter output alternative power have been presented in the literature. This makes the author discuss this variety as they are in the studies with their formulas.  

For the table on page 9, which becomes Table 4, all the undefined acronyms have been listed in a new additional table to define the climate classification symbols in the new version of the manuscript.

  1. Equations (2) and (3) conflict to the reference to them.

The authors are agreed with the reviewer’s comment about Equations (2) and (3), where the reference has been modified accordingly in the new version of the manuscript.

  1. In section 2.2.1, half of the description is in text, half is in a table. The principle of separation is not clear.

The text or the paragraphs in Section 2.2.1 show the guidelines or inverter manufacturers’ recommendations based on the PV sizing ratio, while Table 1, which becomes Table 2 in the revised version, summarizes briefly the recommendations of some PV manufacturers and academics as a concrete example in the commercial markets. However, the following paragraph has been added at the beginning of this section:

“The content of this section can be divided into three; the first part discusses the guidelines or inverter manufacturers’ recommendations based on the PV sizing ratio, while the table summarizes briefly the recommendations of some PV manufacturers and academics as a concrete example in the commercial markets, and finally, a graphical representation is presented for the chronological summary of the main PV-Inverter ratio sizing studies.”

  1. Where did the optimal values come from in Table 2 and why they differ from the recommended ones is not clear.

A new column including the method of the formula of calculating the optimal sizing ratio has been involved in Table 2, which becomes Table 4 in the new version of the manuscript.

The optimal sizing values differ from the recommended ones because The recommended size ratio considered all power losses that would affect the inverter's power generation and conversion efficiency when it was in use.

The new version of the manuscript modified a paragraph accordingly into the following:

. “The optimal sizing ratio, according to Burger et al. [15], relies on the geographic location characteristics, the PV inverter, and the module material composition. To reduce the influence on calculating the chosen inverter size and to prevent errors in power distribution, the study recommended that the precision of the measured time interval must not be below five minutes or fewer. In contrast, the recommended size ratio took into consideration all power losses that would affect the inverter's power generation and conversion efficiency when it was in use.” 

  1. Section 3 is a kind of "protocol of intent" and contains neither the results nor the analysis and characteristics of the initial data with the procedures for their selection, which is a significant part of machine learning methods.

Thanks to the respective reviewer for this comment. More details about the concerned part were added, where a new subsection (3.3 results) has been explored with its figures and table to provide analysis and verification for the proposed optimization method.

  1. The conclusions are abstract, they do not contain useful information.

The authors agree with the reviewer’s comment about the conclusions. The conclusion section has been modified accordingly in the revised version of the manuscript.

Reviewer 2 Report

I found an important review in the paper's subject theme. However, the authors have purposed a strategy to minimize the yearly PV average energy based on Deep Learning networks, and to me, it's only an idea without a test or validation. To improve the paper's quality, I suggest including at least two studies cases or removing the proposed idea and let only the review theme.

Author Response

I found an important review in the paper's subject theme. However, the authors have purposed a strategy to minimize the yearly PV average energy based on Deep Learning networks, and to me, it's only an idea without a test or validation. To improve the paper's quality, I suggest including at least two studies cases or removing the proposed idea and let only the review theme.

Initially, I’d like to thank the reviewer for his time and comment that certainly will contribute to improving the paper's quality. 

Most of the respective reviewer comments have been considered in the revised version of the manuscript. Only I'd like to apologize for not removing the proposed Deep-Learning-based technique, which has been improved by adding more model descriptions and main results as well as that there is another reviewer who asked to improve as well.

Thanks for understanding 

We hope that the reviewers will find our responses to their comments satisfactory, and we are willing to finish the revised version of the manuscript including any further suggestions that the reviewers may have.

Reviewer 3 Report

This paper is a review paper which aims to provide a further insight into the optimization of the PV-Inverter ratio in grid-tied PV systems. The topic is still timely although it has been reviewed already in several papers. It seems like the study is focused on analyzing the PV technologies and the different sizing rations proposed for each technology. Is this the point the review is trying to address? In addition, the authors provide a further insight on the recommendations by each manufacturer. There are some points missing:

1.        It is recommended that the authors include why the review they conducted is more thorough compared to previously published review papers such as [A], [B]. What are the specific gaps that need to be further investigated? Introducing deep learning in this issue which specific issues would resolve? Why are analytical studies like [C] not appropriate?

2.        Several issues are not studied, for example, is there a specific sizing for tracking systems, such as [D]? The authors also mention about sizing when storage systems are involved. Have they considered the sizing when reactive power is provided? This reactive power provision increases the losses as stated in [E]. A comprehensive review on this topic exists in Section 2.2 of [F].

3.        A categorization among analytical methods (reviewed in [F]), optimization packages and deterministic methods based on case studies would be also a good approach.

Generally, the review should be more specific on the added value of this review paper compared to similar ones. Moreover, the reviewer suggests that a discussion section should be included to highlight the specific gaps in the proposed approaches. Is it the computational burden? A minor issue are the superscripts and subscripts in the document.

[A] M. Karimi, H. Mokhlis, K. Naidu, S. Uddin, A.H.A. Bakar “Photovoltaic penetration issues and impacts in distribution network – A review”, Renew. Sust. Energ. Rev., vol. 53, pp. 594–605, 2016.

[B] T. Khatib, A. Mohamed and K. Sopian, “A review of photovoltaic systems size optimization techniques,” Renewable and Sustainable Energy Reviews, vol. 22, pp. 454–465, Jun. 2013

[C] C. S. Demoulias, “A new simple analytical method for calculating the optimum inverter size in grid-connected PV plants”, Electric Power Systems Research, vol. 80, no.10, pp. 1197–1204, 2010.

[D] Kyriaki-Nefeli D. Malamaki and Charis S. Demoulias., “Minimization of electrical losses in two - axis tracking PV systems”, IEEE Transactions on Power Delivery, vol. 28, no.4, p. 2445 – 2455, Oct. 2013.

[E] K. D. Malamaki and C. S. Demoulias, "Estimation of Additional PV Converter Losses Operating Under PF ≠ 1 Based on Manufacturer's Data at PF = 1," IEEE Transactions on Energy Conversion, vol. 34, no. 1, pp. 540-553, March 2019

[F] K. N. Malamaki, PhD Thesis, “Analytical and Detailed Evaluation of Power and Energy Losses in PV Plants aiming at their Integration in Future Ancillary Services Markets” https://ikee.lib.auth.gr/record/332342/files/GRI-2021-31027.pdf

Author Response

This paper is a review paper which aims to provide a further insight into the optimization of the PV-Inverter ratio in grid-tied PV systems. The topic is still timely although it has been reviewed already in several papers. It seems like the study is focused on analyzing the PV technologies and the different sizing rations proposed for each technology. Is this the point the review is trying to address? In addition, the authors provide a further insight on the recommendations by each manufacturer. There are some points missing:

Initially, I’d like to thank the reviewer for his time and comment that certainly will contribute to improving the paper's quality.

This study focuses on the issues of different PV component sizing methodologies including the PV-inverter power sizing ratio, and recommendations for PV-inverter systems by summarizing the power sizing ratio, related derating factor, and sizing formulae approaches. In addition, the presented study recommended a Deep-Learning-based technique that might provide fully automatic computation for the DC/AC sizing ratio and effectively lower the whole return on investment (ROI) over a variety of circumstances and climatic changes.

The above paragraph was added at the end of the introduction section of the new version of the manuscript.

  1. It is recommended that the authors include why the review they conducted is more thorough compared to previously published review papers such as [A], [B]. What are the specific gaps that need to be further investigated? Introducing deep learning in this issue which specific issues would resolve? Why are analytical studies like [C] not appropriate?

The authors thank the respective reviewer for his recommendations.

Study [A] discussed the issues affecting the distribution system as a result of PV penetration such as harmonics, voltage balance, voltage rise, and voltage fluctuation, and their consequence on the system. However, this study didn’t discuss the PV-inverter power sizing ratio.

Although the paper [B] reviewed the sizing optimization issues of PV systems and took into account grid-connected systems, hybrid PV/wind/diesel generator systems, hybrid PV/wind systems, hybrid PV/diesel generator systems, as well as standalone PV systems, this study didn’t discuss the sizing optimization problems for inverter under-sizing, as well as it becomes quite old to the inverter sizing ratio technology, where it reviewed over 100 articles in the period of (1982–2012)

Analytical studies like [C] calculated the optimum inverter size in grid-tie PV systems but with limited (four) unidentified parameters, one is related to the location, and three of which are related to the inverter. In the same context, the optimal inverter size for PV systems placed on two-axis tracking mechanisms in European locales is estimated analytically in [D]. The duration curve of the power at the PV array's dc terminals is used as the foundation for the analytical formulation of the ideal inverter size. However, inverter under-sizing issues and inverter clipping have not been taken into consideration and the calculations were constrained by the inverter's maximum output power.

  1. Several issues are not studied, for example, is there a specific sizing for tracking systems, such as [D]? The authors also mention about sizing when storage systems are involved. Have they considered the sizing when reactive power is provided? This reactive power provision increases the losses as stated in [E]. A comprehensive review on this topic exists in Section 2.2 of [F].

As in the previous reply for the study [C], the optimal inverter size for PV systems positioned on two-axis tracking mechanisms in European locales was estimated also analytically in [D]. The duration curve of the power at the PV array's dc terminals was used as the foundation for the analytical formulation of the ideal inverter size. However, inverter under-sizing and inverter clipping issues have not been taken into consideration and the calculations were only constrained by the inverter's maximum output power.

Overvoltage issues are frequently brought on by the growing use of photovoltaic (PV) systems in distribution networks. The provision of reactive power (RP) by the PV converters is one approach to solving this problem. Increased power losses on the PV converters as a result could raise operating expenses. This issue has been discussed by [E], where the losses are individually computed for the system inverter as the losses are affected by the RP. These losses are comprehensively reviewed in section 2.2 of [F]

  1. A categorization among analytical methods (reviewed in [F]), optimization packages and deterministic methods based on case studies would be also a good approach.

Thanks to the respective reviewer for this valuable comment.

An additional subsection entitled “2.3. Analytical methods affect inverter in PV inverter” has been added accordingly. The above fundamental aspects have been mostly discussed in this new section of the revised version of the manuscript.

Generally, the review should be more specific on the added value of this review paper compared to similar ones. Moreover, the reviewer suggests that a discussion section should be included to highlight the specific gaps in the proposed approaches. Is it the computational burden? A minor issue are the superscripts and subscripts in the document.

A comparison with similar review work on DC/AC sizing for PV-inverter systems has been added in the new version of the manuscript accordingly.

[A] M. Karimi, H. Mokhlis, K. Naidu, S. Uddin, A.H.A. Bakar “Photovoltaic penetration issues and impacts in distribution network – A review”, Renew. Sust. Energ. Rev., vol. 53, pp. 594–605, 2016.

[B] T. Khatib, A. Mohamed and K. Sopian, “A review of photovoltaic systems size optimization techniques,” Renewable and Sustainable Energy Reviews, vol. 22, pp. 454–465, Jun. 2013

[C] C. S. Demoulias, “A new simple analytical method for calculating the optimum inverter size in grid-connected PV plants”, Electric Power Systems Research, vol. 80, no.10, pp. 1197–1204, 2010.

[D] Kyriaki-Nefeli D. Malamaki and Charis S. Demoulias., “Minimization of electrical losses in two - axis tracking PV systems”, IEEE Transactions on Power Delivery, vol. 28, no.4, p. 2445 – 2455, Oct. 2013.

[E] K. D. Malamaki and C. S. Demoulias, "Estimation of Additional PV Converter Losses Operating Under PF ≠ 1 Based on Manufacturer's Data at PF = 1," IEEE Transactions on Energy Conversion, vol. 34, no. 1, pp. 540-553, March 2019

[F] K. N. Malamaki, PhD Thesis, “Analytical and Detailed Evaluation of Power and Energy Losses in PV Plants aiming at their Integration in Future Ancillary Services Markets” https://ikee.lib.auth.gr/record/332342/files/GRI-2021-31027.pdf

Round 2

Reviewer 1 Report

I have no new comments

Reviewer 2 Report

The effort to improve the article is appreciated, I can see that the authors have made important improvements to the document, enriching themselves with evidence of the proposal described in the text.

Reviewer 3 Report

The quality of the manuscript has been improved. Please check spelling and syntax.